# Smartphone App Usage Patterns for Trip Planning Purposes and Stated Impacts in the City of Bhopal, India

**Kushagra Sinha *** and Sanjay Gupta

Department of Transport Planning, School of Planning and Architecture, Block-B, Indraprastha Estate, New Delhi 110002, India
* Correspondence: kush.spab@gmail.com

**Abstract:** With the considerable growth in the information and communication technology (ICT), several smartphone-based mobility platforms have already sprung up and they have the potential of transforming the mobility ecosystem completely. However, there is close to no knowledge available about how ICT-based smartphone apps meant for day-to-day trip planning tasks are being used across various user groups and how they influence travel outcomes, especially in Indian cities. Therefore, this study is an effort to close this gap by gathering data from the city of Bhopal and carrying out an exploratory statistical analysis on the usage of smartphone apps for different types of trip planning purposes, as well as their influence on travel outcomes. The study provides empirical evidence of relationships between smartphone app usage for trip planning (such as departure time, destination selection, mode selection, route selection, communicating and coordinating trips, and performing tasks online rather than visiting) and the resulting travel outcomes, such as kilometres travelled by vehicle (for purposes, such as work/education, shopping, and recreation), social gatherings, new destinations, and group trips. The chi-square test has been used to test and interpret several socioeconomic variables that could influence this relationship, such as gender, age, income, etc. The study's findings provide important behavioural insights that may be useful in policy discussions.

**Keywords:** smartphone apps; trip planning; app usage; travel outcomes





## 1. Introduction

Information and communication technologies (ICTs) are becoming more readily accessible, particularly in the form of mobile phones, which have changed people's lifestyles: how they live, work, shop, and travel. Most of these changes are happening due to the availability of different options for destinations, mode, route, and travel time made available through these devices [1]. For instance, users frequently use smartphone applications, or "apps", for a number of transportation use cases. These apps offer a variety of advanced features, including real-time location-specific data. They offer details on destinations as well as travel options to those destinations. These apps allow users to access information, such as travel costs, transit schedules, travel routes, required travel times, and fastest modes. More individuals are starting their journeys with their smartphones to plan out their routes, check the schedules of the upcoming buses, trains, or metros, hail a cab, or locate a private driver utilising the services offered by app-based cab aggregators. Use of these ICT related services results in the reorganization of activities [2] and thus associated travel patterns are also changing owing to improvements in the efficiency of travel [3].

Existing research on the subject indicates that smartphones are growing in popularity among users of the younger age [4]. It has been established that more empirical studies are required to confirm the association between smartphone use and travel behaviour in general [5]. ICT's influence on travel behaviours can be shaped by a number of factors. Individual characteristics [6], such as trip frequencies [7], e- and tele-shopping [7,8], frequency of the use of the internet, mobile phone ownership by a household, telephones at home or

ones used for business purposes [9,10], and personal computers are a few examples [11]. However, it is still unclear how using a smartphone affects how people organise their regular trips.

To the best of the authors' knowledge, limited study has been carried out in this area. A future study agenda on the impacts of ICT on travel behaviour was outlined by Van Wee, Chorus, and Geurs (2013) who identified a number of relevant elements [12]. They observed that there is still a gap in the literature regarding research on the relationship between ICT and travel comfort. The study also covered how mobile phone technologies affect people's access to various types of information. Additionally, they argued that the growing use of smartphones will cause ICT to play an important role in trip planning. Based on this, Jamal and Habib (2019) conducted a study to examine the empirical evidence of associations between smartphone app use for trip planning (such as departure time, trip destination, transport mode choice, communication/trip coordination, and performing various daily tasks online) and subsequent travel outcomes, such as kilometres travelled by vehicle, social gatherings, new destinations, and group travel, as well as factors associated with them [13], and they concluded that younger people's travel outcomes are mostly influenced by smartphone apps. The study also showed that the use of smartphone applications has a limited influence on lowering travel outcomes, such as car miles travelled, with new place exploration having the biggest effects. Although such studies could potentially serve as a foundation for subsequent research into the extent to which different factors of smartphone app usage have an impact on trip planning and travel outcomes, they have not been undertaken in the context of Indian cities.

In India, there are 85 mobile phone connections for every 100 people, and during the past ten years, mobile phone use has increased significantly [14]. As previously discussed, mobile phones have evolved into powerful information, communication, sensing, and entertainment gadgets known as smartphones, and nearly 24% of Indians own smartphones [15]. Although slower than many emerging economies, smartphone ownership has been growing at a healthy rate [16], mainly due to the push provided by the Government of India towards creating digital infrastructure and digital literacy through the Digital India initiative which aims to facilitate the high-paced penetration of mobile connectivity in large urban centres, medium to smaller towns, and distant villages alike [17]. Given that smartphones are used to access the internet by 81% of internet users in India, these devices are a vital component of internet penetration [18]. Additionally, to create 100 sustainable and liveable cities, the Indian government presented its flagship initiative, the "Smart Cities Mission," in 2015. These cities aim to combine and use the latest technologies available to create a technologically advanced and highly connected urban infrastructure [19]. Owing to this, the work on smartphone-enabled mobility platforms, similar to mobility as a service (MaaS) has already begun in India. Together with other key stakeholders, the Ministry of Housing and Urban Affairs (MoHUA), the Government of India is developing a framework to introduce them in Indian cities for integrating all shared transportation through a single app with numerous services related to trip planning and payment [20]. Thus, it is not going to be long before such platforms start emerging, beginning in larger cities, and then eventually penetrating the medium size cities as well.

However, as mentioned before, the foundational research on transport app usage and its influence on travel, which is critical to be considered for such deployments, are not available in the context of Indian cities. It presents a major challenge because such deployments also require a lot of knowledge sharing and possibly, transfer of technology from the developed nations where such platforms are already being tested/deployed and the similarities or the dissimilarities in the app usage, and its influence on travel outcomes between cities of developed countries and India are required to be established. It is especially important to study this matter in the context of medium-sized Indian cities because for most residents in such cities, using smartphone apps for travel requirements is a relatively novel trend, and the knowledge of users' usage patterns of existing apps is unavailable. The socio-economic factors driving the app usage are also unknown, and it is

unclear what kind of users are adopting them. Additionally, it is unknown if the use of these smartphone apps has altered how people move about Indian cities.

As a result, this study is an effort to close the gap by gathering data and carrying out exploratory statistical analysis on the usage of smartphone apps for different types of trip planning purposes, as well as their influence on travel outcomes. A variety of trip planning purposes have been considered in this study, including departure time, trip destination, transport mode choice, communication/trip coordination, and performing various daily tasks online. The stated changes in kilometres travelled by vehicle (for work/education, shopping, and recreation), number of new places visited, social events attended, and planned group trips have been undertaken as travel outcomes. The sections that follow begin with a brief review of the body of literature on how ICT affects travel decisions and outcomes. The paper then presents the survey data and method of analysis, followed by a discussion of the results/findings. Finally, an overview of the research and directions for the future are included in the paper's conclusion.

## 2. Literature Review

ICT's influence on travel has been well explored in the literature, particularly in developed countries, and these studies have mostly focussed on how ICT can either increase or decrease travel [11,21–26]. However, this study varies from most others because it primarily investigates the inherent relationship between the use of smartphone apps and day to day behaviours associated with travel. In other words, the study investigates how heavily individuals rely on smartphone apps for trip planning purposes and as a result of this reliance, what changes they are noticing in their travel behaviour related outcomes. Travel-related studies mainly consider the impact of usage of all types of relevant ICTs, including telecommunication devices (such as mobile phones and landlines) and personal computers, but there are relatively fewer studies specifically looking at the effects of smart phones on travel even though smartphones are already capable of performing the majority of functionalities of most ICT devices.

A study in the Osaka metropolitan area of Japan investigated the relationships between telecommunication (such as mobile phones and landlines) and activities and found that the usage of communication technology decreases work-related activities, enhances leisure activities, and has no effect on maintenance tasks [27]. Numerous studies investigated how sociodemographic factors affected people's use of ICT. Most internet users are young and middle-aged persons, along with students [13,28]. Most mobile phone/smartphone users [4,11] likewise, belonged to the same age groups. Men used the internet more frequently than women [28]. However, a US survey of St. Louis metro users found that smartphone ownership is gender neutral [4]. Participation in online activities also rises along with income level [28]. Bhat, Sivakumar, and Axhausen (2003) and Mondschein (2011) found a similar pattern, demonstrating an increased probability of owning a cell phone rises with income [11,29]. However, as per Srinivas and Athuru (2004) the ownership of smartphones was found to be less significantly associated with income [4].

A survey of US travellers found that online planning and the purchasing of travel-related products and services, such as hotel rooms and tickets, has significantly increased in recent years [30]. Srinivasan and Athuru (2004) investigated the engagement in virtual activities (using the internet), such as online banking and maintenance and discretionary activities in the San Francisco Bay area [28] and their study found that while using the internet shortens journeys, day to day maintenance-related activities and trip-making frequency increase. Corpuz and Peachman (2003) showed that internet use has a significant influence on personal transactions (such as banking) and travel for educational purposes [31]. According to Bhat, Sivakumar, and Axhausen (2003), individual trips made for non-maintenance activities, such as shopping, is declining because of the increasing usage of mobile phones and personal computers [11]. Conversely, it has also been found that in some cases, the frequency of shopping trips has actually increased with rise of online shopping [7,32]. It

has also been observed that females and individuals belonging to older age groups are less engaged in online shopping [7].

Some research pertaining to the inherent relationships between trip characteristics and travel outcomes are also available. A study conducted by Wang and Law (2007) in Hong Kong showed that using ICT-based services, such as email, the internet, video calling and conferencing, etc., enhances recreational activities, the propensity for trip-making, and travel time [33]. Internet use was revealed to be negatively correlated with travel time; however mobile phone use was positively correlated [34]. Hjorthol (2002) observed that, after adjusting for socio-economic factors, such as gender, age, income, and car ownership, using a personal computer for work both with and without an internet connection has a slight but statistically significant positive impact on distance travelled and overall trips on a daily basis [35]. A Chicago study discovered a positive association between social travel and cell phone use [29]. In the Netherlands, ICT use had a neutral or substitutive impact on social travel, while internet interactions had a negative impact on social travel distance [36]. However, Carrasco (2011) found complementary effects of ICT devices in Chile [37]. Wang and Fesenmaier (2013) investigated how using a smartphone for entertainment, communication, convenience, and information searches affected how well a trip went [38]. According to the study, a smartphone can change how travellers choose to travel, how they organise their trips, and how easily they can access information.

Most of the studies in the current literature, which involve developed countries, focused on the broad effects of ICT on travel. There is little evidence, of how ICT-based smartphone apps meant for regular trip planning tasks influence travel outcomes. Jamal and Habib (2019) used a chi-square test to assess the variability of app usage for various trip planning purposes, such as choosing to complete tasks online instead of travelling, communicating/coordinating trips, and selecting travel destinations, modes of transportation, and departure times, and stated changes in travel outcomes, such as Vehicle Kilometres Travelled (VKT), as well as other outcomes, such as the number of new places visited, social gatherings attended, and planned group trips [13]. The study found that the majority of users use smartphone apps for the purpose of communication and coordination. They also found that using smartphone apps has a limited influence on lowering travel outcomes, such as car miles travelled, with new places explored having the biggest effects.

There has not been any equivalent research on Indian cities, to the extent that the authors are aware. In order to address this gap, a survey of smartphone users was conducted in the city of Bhopal (India). Additionally, the study provides an extensive exploratory analysis of the primary survey data on the use of smartphone apps for day-to-day trip planning activities and their stated impacts on travel outcomes.

## 3. Materials and Methods

### 3.1. Study Area

The Indian city of Bhopal has been chosen as the case study for this research. It is the capital of the state (province) of Madhya Pradesh and is in Central India. The city is a hub of knowledge with many public and private universities, culturally vibrant neighbourhoods with long histories, and various man-made and natural lakes. As per the census of India (2011), more than two million people [39] reside within the 813 sq. km (with a density of 2482 persons/sq.km) of the urban agglomeration of Bhopal (including Kolar area) and, as per the trend-based estimation, it was expected to have grown to about 2.5 million by 2022. The per capita net income in Madhya Pradesh increased to INR 103,288 or USD 1656.58 [40], putting it in 13th place among the other states. So relatively, it can be considered a medium-income state, and the per-capita income for the Bhopal urban agglomeration has been estimated to be INR 134,982 for the year 2022, using a trend-based analysis of past data [41]. The city also has an average literacy rate of 83.47%.

Bhopal has compelling reasons to be considered as a case study. It was picked in the first round for the 20 lighthouse cities, out of the 100 cities chosen for the Smart Cities Mission, the purpose of which is to turn the city into a smart, sustainable, and liveable

city that is prepared for the future. A 186-kilometre bus rapid transit system (BRTS) is operational in the city and is integrated with an app called 'Chalo' which provides real-time information about the bus service. A tap-to-pay bus card named "Chalo card" has also been introduced, which has a prepaid wallet and the capability to store bus passes. The public bicycle sharing (PBS) system has been deployed on multiple stretches in the city, primarily in the catchment of the BRTS. App-based shared mobility platforms, such as Uber, Ola, and Rapido have already provided their services. Construction work on Phase 1 of the Bhopal Metro commenced in January 2019 and is expected to be completed by 2026. So, with almost all of the modern forms of mobility systems (both existing and committed) and increasing smartphone penetration, the city already has the ingredients for deploying MaaS, and as the city grows in size and complexities, trip planning will become a critical part of everyday life for the residents. Many such apps are already available in the city (Table 1).

**Table 1.** Available smartphone apps for trip planning purposes in Bhopal.

| Trip Planning Purposes→ <br><br> Smartphone Apps Available in Bhopal | Deciding Departure Time | Deciding Destination | Mode Choice | Route Selection | Communication and Coordination | Online Tasks |
|---|---|---|---|---|---|---|
| Map and Navigation Services (e.g., Google Maps and Apple Maps) | ✓ | | ✓ | ✓ | ✓ | |
| Public Transport Apps (e.g., Chalo App) | ✓ | | ✓ | | | |
| Shared Mobility Apps (e.g., Uber, Ola, InDriver, Rapido, Chartered Bikes, etc.) | ✓ | | ✓ | | | |
| Information Apps for Recreational Activities (e.g., BookMyShow, Zomato, etc.) | ✓ | ✓ | | | | ✓ |
| Ticketing and Payment Apps (e.g., PayTM, PhonePay, Bharat Pay, GPay, etc.) | | | ✓ | | | ✓ |
| Social Network Apps (e.g., Facebook, WhatsApp, Twitter, etc.) | | ✓ | | ✓ | ✓ | |

*3.2. Data Collection*

The primary data for this study were collected through an online survey of smartphone users in Bhopal between September and December of 2021. To reach the respondents, a survey invitation was circulated through platforms, such as WhatsApp, Facebook, and Instagram. Personal level details, such as residential location, gender (male and female), age group (users below 18 years of age were not been considered in the study as their mobility is highly dependent on others), and years of smartphone use; and household level details, including household composition (with or without children below 18 years of age), monthly household income, four-wheeler ownership, and two-wheeler ownership were recorded as categorical choices. The survey gathered data related to the usage of smartphone apps for making travel-related decisions and the stated changes in travel outcomes, among other things. Respondents were asked questions about their use of smartphone apps for trip planning purposes, including deciding when to depart for the trip, deciding on the trip destination, choosing a mode of transportation, communicating, and coordinating trips with others, and performing tasks online rather than travelling to a location. A 5-point Likert scale was used to collect the data, with options such as never, rarely, sometimes, often, and always. Questions related to travel outcomes included the influence of smartphone usage on VKT (for work/education, shopping, and recreational trips), the number of new places visited, group trips planned, and social gatherings attended. Again, a 5-point Likert scale was used to collect the data, with options, such as significantly reduced, slightly reduced, no impact, slightly increased, significantly increased.

Through snowball sampling, the online survey produced a sample of 548 smartphone users residing in Bhopal, out of which, 475 completed samples were selected (95 each from every age group under consideration in the study). Table 2 shows the demographic distribution of the collected samples from the survey.

**Table 2.** Socio-Demographic distribution of the selected survey respondents.

| Socio-Demographic Variables | | Percentage |
|---|---|---|
| Gender | Male | 52% |
| | Female | 48% |
| Age Group (in Years) | 18–24 | 23% |
| | 25–34 | 25% |
| | 35–44 | 21% |
| | 45–54 | 14% |
| | 55–64 | 8% |
| | 65 Years and above | 8% |
| Education Level | High | 36% |
| | Medium | 23% |
| | Low | 41% |
| Years of Smartphone Use | Less than 1 | 2% |
| | 1–3 | 12% |
| | 3–5 | 18% |
| | More than 5 | 68% |
| Household Composition | With children under 18 Years | 54% |
| | Without children under 18 Years | 46% |
| Monthly Household Income (in INR) | Less than 5000 | 20% |
| | 5000–20,000 | 20% |
| | 20,000–50,000 | 20% |
| | 50,000–100,000 | 20% |
| | More than 100,000 | 20% |
| Four-wheeler Ownership | None | 42% |
| | One | 34% |
| | Two | 17% |
| | Three or More | 7% |
| Two-wheeler Ownership | None | 27% |
| | One | 41% |
| | Two | 27% |
| | Three or More | 6% |

### *3.3. Data Analysis Method*

In order to study the app usage pattern, the responses were first converted into a numerical scale, with never = 1, rarely = 2, sometimes = 3, often = 4, and always = 5. Then, these mean scores were used to create graphical plots to better understand the app usage patterns. A higher mean score for an app user category indicates a greater reliance on smartphone apps. Further, the relationship between smartphone app usage for trip planning purposes and socio-demographic indicators (as mentioned previously) was tested for establishing how the use of smartphone applications varies among several groups. A chi-square test was performed to assess the existence of an inherent relationship between variables. This statistical test thus assumes a null hypothesis—*There is no relationship between smartphone app usage for trip planning purposes and socio-demographic variables considered.*

Finally, this study also investigated the stated impact of smartphone app usage on travel outcomes. Again, a chi-square test was performed, and it assumed a null hypothesis—*There is no relationship between stated impacts on travel outcomes and socio-demographic variables considered.*

The results of the test, along with the pattern of app usage for trip planning purposes and the stated impacts on travel outcomes in Bhopal have been presented as per socio-demographic classifications followed by a discussion of the key findings, both novel and ones similar to existing studies.

## 4. Results

The results in this section have been organized into two parts. First, the smartphone app usage pattern for trip planning purposes across various user groups based on personal and household attributes has been discussed. It is then followed by the assessment of the stated changes in travel outcomes because of smartphone app usage for trip planning purposes.

*4.1. Smartphone App Usage Pattern*

The pattern of use of smartphone apps for trip planning purposes by various user groups based on personal level attributes is shown in Figure 1.

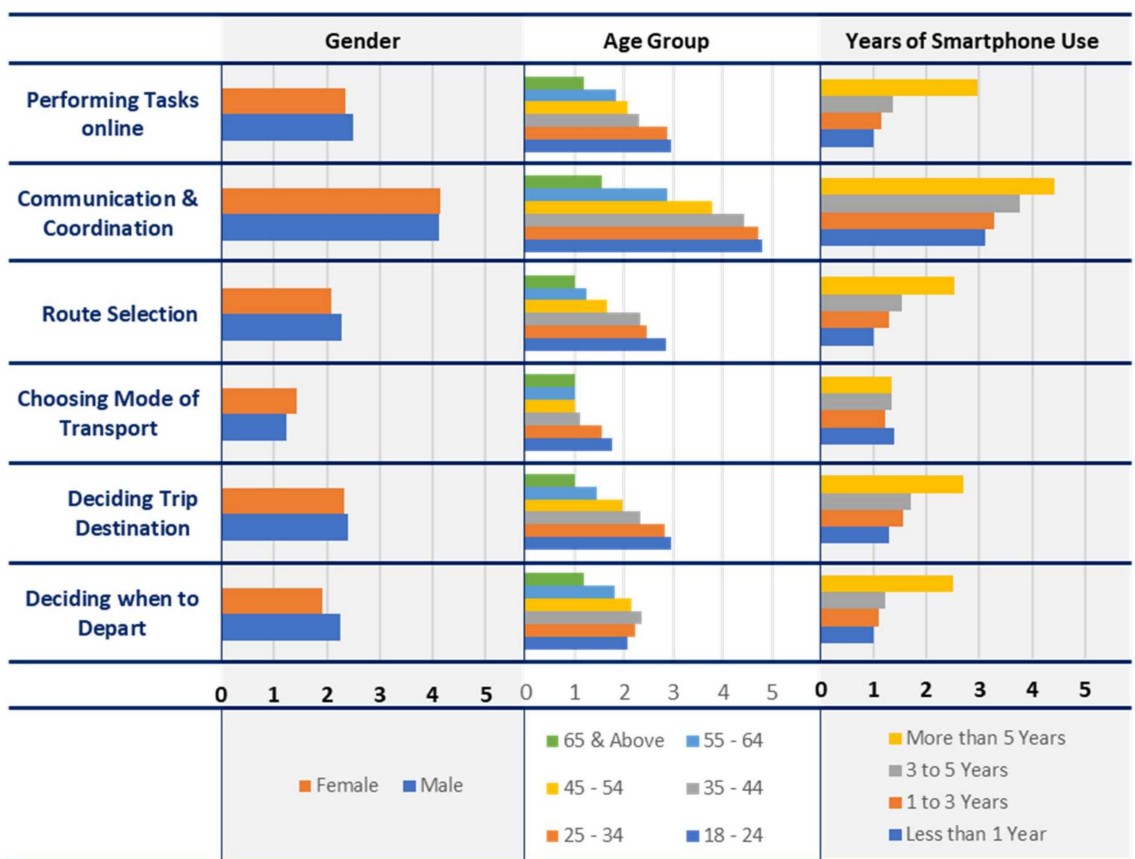

**Figure 1.** Personal level mean scores for trip planning purposes.

Overall, except for communication and coordination enabled by navigation services and social networking apps, users across gender, age and years of smartphone usage show a lower dependence on smartphone apps. Choosing the mode of transport is the least preferred use. The mean score for various trip planning purposes across gender is relatively similar for most purposes. Young users aged 18 to 44 have a higher score than older users, showing higher usage of apps for most purposes, except for the purpose of deciding when to depart for which, users between ages 25 to 54 years showed a higher dependence. Users experienced with more than five years of smartphone use show higher mean scores for all purposes. The chi-square test has been used to investigate the association between personal level user attributes (gender, age group, and years of smartphone use) and app usage patterns for trip planning purposes. For example, in the case of gender, the test seeks

to check if the difference in app usage patterns between male and female respondents exists due to an inherent relationship between gender and app usage patterns. The *p*-values for relationships that are less than the chi-square critical value (5% significance level; $p < 0.05$) have been considered significant and null hypotheses for them have been rejected (Table 3), thus showing that there is a significant relationship.

**Table 3.** Estimated levels of significance (*p*-values) for the tested relationship between personal and household level socio-demographic attributes and app usage for trip planning purposes.

| Purpose of App Use | Personal Level | | | Household Level | | | |
| --- | --- | --- | --- | --- | --- | --- | --- |
| | | | | | | Vehicle Ownership | |
| | Gender | Age Group | Smartphone Use (Years) | Household Composition | Household Income | Four-Wheeler | Two-Wheeler |
| 1. Deciding when to Depart | 0.001 | 0.001 | 0.000 | 0.049 | 0.000 | 0.000 | 0.000 |
| 2. Deciding Trip Destination | 0.369 * | 0.000 | 0.000 | 0.070 * | 0.000 | 0.000 | 0.000 |
| 3. Choosing Mode of Transport | 0.000 | 0.000 | 0.000 | 0.504 * | 0.000 | 0.015 | 0.023 |
| 4. Route Selection | 0.556 * | 0.000 | 0.000 | 0.396 * | 0.000 | 0.000 | 0.000 |
| 5. Communication and Coordination | 0.844 * | 0.000 | 0.000 | 0.966 * | 0.000 | 0.000 | 0.000 |
| 6. Performing Tasks Online | 0.153 * | 0.000 | 0.000 | 0.300 * | 0.000 | 0.000 | 0.000 |

* Null hypothesis is accepted.

At the personal level, the test rejects the null hypothesis for gender for trip purposes, such as deciding when to depart and choosing a mode of transport, where a difference in app usage patterns has been observed between the male and female respondents due to an inherent relationship between gender and mentioned trip planning purposes. For both purposes, it can be observed that females are slightly less likely to use smartphone apps. For age group and years of smartphone use for all purposes of trip planning, the test also rejects the null hypothesis, demonstrating a significant relationship. Age has been negatively related to smartphone app usage, and the propensity for app usage keeps decreasing. The younger users are more likely app users for the stated purposes. However, among the younger users, app usage for deciding when to depart and route selection find relatively less dependence. It has also been observed that more years of smartphone ownership and usage increases reliance on app usage. An exception to this is the purpose of communication and coordination, for which it has been observed that even relatively new users show significant reliance on smartphone app usage.

Further, the usage pattern of smartphone apps for trip planning purposes by various user groups based on household-level attributes is shown in Figure 2.

As observed earlier, users mainly use apps for communication and coordination across all household level attributes and choosing the mode of transport is the least preferred use. It has been observed that the mean score of app usage is not much affected by the presence of children under 18 years in the city. However, other user attributes, such as monthly income and vehicle ownership have been observed to show variations in the mean score. The mean score for app usage by high-income users has been observed to be higher for all purposes and steadily decreases for lower-income groups. Communication and coordination are the most preferred purpose for which users use apps among all income categories. Although the use of apps for choosing a mode of transport has the least mean score across all purposes, it has been observed that only the users of medium-income households have a relatively higher usage. It has also been observed that the absence of a private vehicle discourages users from using smartphone apps for the mentioned purposes. Users who own a four-wheeler or a two-wheeler have higher mean scores for app usage. However, in choosing a mode of transport, households with private vehicles show relatively less dependence on smartphone use.

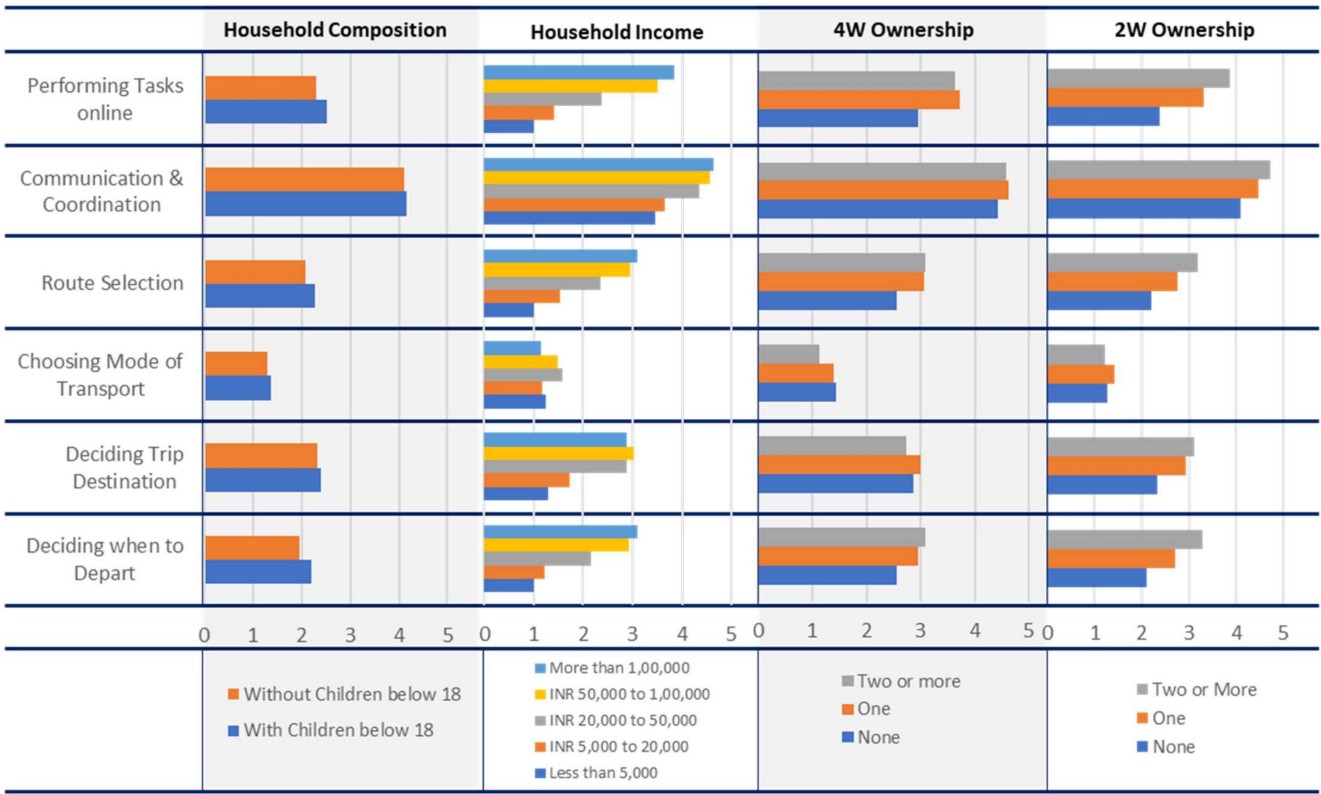

**Figure 2.** Household-level mean scores for trip planning purposes.

The association between user attributes at the household level (such as household composition, income, and vehicle ownership) was investigated using the chi-square test and app usage patterns for trip planning. (Table 2). The chi-square test rejects the null hypothesis for household composition to decide when to depart only, thus showing that a significant difference in the app usage pattern for the mentioned purpose is due to an inherent relationship between household composition and mentioned purpose. For all other trip planning purposes, such a relationship, does not exist. The test also rejects the null hypothesis for monthly household income, vehicle ownership (four-wheeler and two-wheeler) for all trip planning purposes. For most purposes, monthly household income is positively related to app usage, and the higher income of the household encourages it. However, even low-income households rely on smartphone apps for communication and coordination. It has been observed that users who belong to families with at least one private car appear to demonstrate more reliance on smartphone applications for various trip planning purposes. Private ownership of four- and two-wheelers seems to encourage app usage as well.

*4.2. Impact Assessment*

The influence of usage of smartphone apps on stated changes to travel outcomes has been investigated at both personal and household levels. The overall share of responses for changes in travel outcomes is shown in Figure 3.

Interestingly, the impact is meagre as most users across various classifications reported 'No Impact' on travel outcomes. Smartphone app usage has the lowest influence on VKT for work/education trips as 73% of respondents stated, 'no impact'. Contrary to expectations, about 27% stated a slight to a significant increase in VKT for the same. For shopping trips, however, 59% of respondents reported a decrease in VKT, and nobody stated an increase in VKT; 42–50% of users stated no impact of smartphone app usage on the rest of their travel outcomes. Only 27% of respondents stated a decrease in VKT for recreational trips, and 32% reported an increase in VKT for the same. A decrease in social gatherings was

reported by 31% of respondents, while an increase was reported by 21%. A similar decline in the number of newly visited locations was reported by 30% and 29% of respondents, respectively, and the frequency of planned group travel, respectively, whereas 22% of respondents stated an increase in the number of new places visited and 20% stated an increase in the number group trips planned.

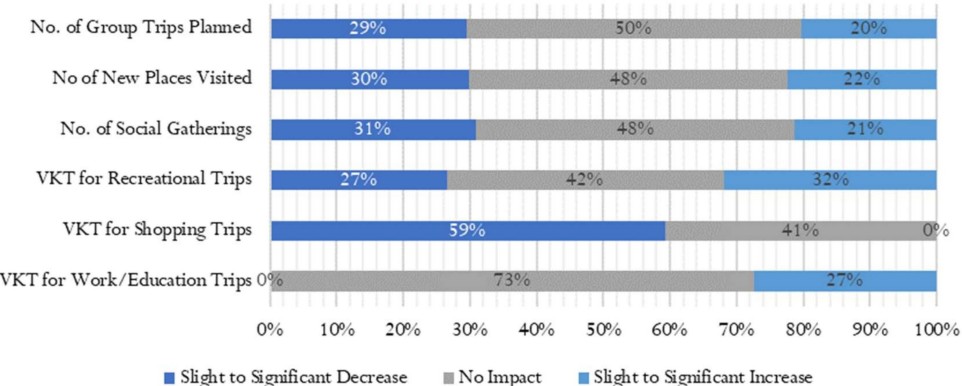

**Figure 3.** Outcome-wise, the share of responses on the impacts of smartphone apps.

Table 4 shows the personal level attribute-wise stated impacts of smartphone apps usage on travel outcomes.

**Table 4.** Personal level attributes and travel outcome-wise classification of smartphone app users.

| Travel Outcome | Stated Impacts | Gender | | Age (in Years) | | | | | | Years of Smartphone Use | | | |
|---|---|---|---|---|---|---|---|---|---|---|---|---|---|
| | | Male | Female | 18–24 | 25–34 | 35–44 | 45–54 | 55–64 | 65 + | <1 | 1 to 3 | 3 to 5 | >5 |
| VKT for Work/Education | Significant Decrease | 0% | 0% | 0% | 0% | 0% | 0% | 0% | 0% | 0% | 0% | 0% | 0% |
| | Slight Decrease | 0% | 0% | 0% | 0% | 0% | 0% | 0% | 0% | 0% | 0% | 0% | 0% |
| | No Impact | 69% | 76% | 48% | 63% | 80% | 86% | 100% | 100% | 100% | 98% | 91% | 63% |
| | Slight Increase | 19% | 18% | 27% | 24% | 20% | 14% | 0% | 0% | 0% | 2% | 8% | 25% |
| | Significant Increase | 12% | 6% | 25% | 13% | 0% | 0% | 0% | 0% | 0% | 0% | 1% | 13% |
| | | *p*-value = 0.053 | | *p*-value = 0.000 | | | | | | *p*-value = 0.000 | | | |
| VKT for Shopping Trips | Significant Decrease | 38% | 34% | 61% | 53% | 24% | 22% | 5% | 0% | 0% | 0% | 15% | 49% |
| | Slight Decrease | 21% | 26% | 19% | 18% | 37% | 20% | 30% | 15% | 0% | 20% | 19% | 26% |
| | No Impact | 41% | 40% | 20% | 28% | 39% | 58% | 65% | 85% | 100% | 80% | 66% | 26% |
| | Slight Increase | 0% | 0% | 0% | 0% | 0% | 0% | 0% | 0% | 0% | 0% | 0% | 0% |
| | Significant Increase | 0% | 0% | 0% | 0% | 0% | 0% | 0% | 0% | 0% | 0% | 0% | 0% |
| | | *p*-value = 0.369 * | | *p*-value = 0.000 | | | | | | *p*-value = 0.000 | | | |
| VKT for Recreational Trips | Significant Decrease | 4% | 6% | 0% | 0% | 0% | 0% | 25% | 30% | 0% | 0% | 1% | 6% |
| | Slight Decrease | 22% | 22% | 15% | 16% | 25% | 29% | 35% | 25% | 40% | 15% | 24% | 22% |
| | No Impact | 41% | 42% | 25% | 24% | 67% | 63% | 40% | 45% | 60% | 85% | 64% | 28% |
| | Slight Increase | 24% | 21% | 48% | 35% | 8% | 8% | 0% | 0% | 0% | 0% | 12% | 30% |
| | Significant Increase | 9% | 9% | 12% | 25% | 0% | 0% | 0% | 0% | 0% | 0% | 0% | 13% |
| | | *p*-value = 0.837 * | | *p*-value = 0.000 | | | | | | *p*-value = 0.000 | | | |
| Number of Social Gatherings | Significant Decrease | 10% | 5% | 15% | 0% | 0% | 0% | 25% | 28% | 30% | 13% | 8% | 6% |
| | Slight Decrease | 22% | 25% | 14% | 23% | 23% | 31% | 28% | 33% | 10% | 15% | 29% | 23% |
| | No Impact | 47% | 49% | 35% | 41% | 59% | 69% | 48% | 40% | 60% | 73% | 58% | 41% |
| | Slight Increase | 16% | 16% | 29% | 22% | 18% | 0% | 0% | 0% | 0% | 0% | 5% | 22% |
| | Significant Increase | 5% | 6% | 7% | 14% | 0% | 0% | 0% | 0% | 0% | 0% | 0% | 8% |
| | | *p*-value = 0.339 * | | *p*-value = 0.000 | | | | | | *p*-value = 0.000 | | | |
| Number of New Places Visited | Significant Decrease | 11% | 8% | 20% | 0% | 0% | 0% | 23% | 35% | 40% | 16% | 8% | 8% |
| | Slight Decrease | 20% | 21% | 10% | 25% | 23% | 22% | 23% | 25% | 10% | 16% | 29% | 19% |
| | No Impact | 47% | 49% | 35% | 38% | 54% | 78% | 55% | 40% | 50% | 67% | 58% | 42% |
| | Slight Increase | 17% | 18% | 28% | 24% | 23% | 0% | 0% | 0% | 0% | 0% | 5% | 24% |
| | Significant Increase | 6% | 3% | 6% | 13% | 0% | 0% | 0% | 0% | 0% | 0% | 0% | 7% |
| | | *p*-value = 0.596 * | | *p*-value = 0.000 | | | | | | *p*-value = 0.000 | | | |
| Number of Group Trips Planned | Significant Decrease | 9% | 9% | 15% | 0% | 0% | 0% | 30% | 38% | 30% | 11% | 7% | 9% |
| | Slight Decrease | 20% | 20% | 15% | 18% | 24% | 29% | 18% | 23% | 10% | 15% | 22% | 21% |
| | No Impact | 48% | 53% | 40% | 46% | 57% | 71% | 53% | 40% | 60% | 75% | 66% | 42% |
| | Slight Increase | 17% | 13% | 23% | 22% | 19% | 0% | 0% | 0% | 0% | 0% | 5% | 20% |
| | Significant Increase | 6% | 5% | 7% | 15% | 0% | 0% | 0% | 0% | 0% | 0% | 0% | 8% |
| | | *p*-value = 0.709 * | | *p*-value = 0.000 | | | | | | *p*-value = 0.000 | | | |

*: Null hypothesis accepted.

It can be observed that among the personal level attributes considered, gender does not have a statistically significant relationship with the travel outcomes for most outcomes, except for a weak relationship with VKT for work/education (as per a *p*-value of 0.05 or less for the chi-square test), where there is a slight difference in the app usage pattern between male and female users. Thus, the difference in the stated impact of app usage on VKT for work/education between male and female respondents, is due to an inherent relationship between gender and mentioned travel outcome (VKT for work/education). Age (and years of smartphone use) was found to have a significant relationship with the stated impacts of app usage for all travel outcomes, with all travel outcomes showing that respondents of different age groups reported differences in their respective travel outcomes due to this relationship. Impacts of smartphone app usage on travel outcomes across personal level attributes for statistically significant relations are discussed below:

- *Gender*: The majority of respondent reported no decrease in VKT for work/education. A larger proportion of female (76%) respondents stated that their commute to work/education was not impacted by smartphone app usage, as compared to male respondents (69%) and thus, a larger proportion of male respondents (31%) reported a slight to a significant increase in VKT for work/education, as compared to females (24%). For the other outcomes, the results are similar for both male and female respondents. Most respondents across genders reported a slight to significant decrease in VKT for shopping and a slightly larger proportion reported a slight to significant increase in VKT for recreational trips. As for other travel outcomes, a slightly larger proportion reported a slight to significant decrease in the number of social gatherings, number of new places visited, and number of group trips planned.
- *Age*: All respondents over 55 years of age, most users between ages 35–54 years (80–86%), and a significantly large number of users between ages 18–34 years (48–63%) stated no impact on VKT for work/education due to smartphone app usage. None of the respondents stated a decrease in VKT for the same. However, 52% and 37% of respondents from 18–24 years and 25–34 years, respectively, stated a slight to a significant increase in VKT for work/education. Respondents belonging to middle age groups of 35–44 years and 44–55 years stated a slight increase in VKT for the same. In the case of VKT for shopping trips, the responses are different. Although still very significant, a comparatively lower number of respondents reported no impact on VKT for shopping trips, and it was observed that as age increased, respondents reported less dependence on smartphone apps for shopping. It is also evident from the observation that most respondents from younger (18–34 years) to early middle (35–44 years) age groups stated a slight to significant decrease in VKT for shopping trips because of app usage. None of these users reported an increase in VKT for shopping trips. It is interesting to note that while most users from middle age groups (35 to 54 years) reported no impact, some reported a slight decrease in the number of group trips planned, the number of social gatherings attended, and the number of new places visited as a result of using smartphone apps. A very few reported a slight increase. In comparison, younger and older age groups reported slight to significant changes. Most respondents from younger age groups (18–34 years) reported a slight to a significant increase, and most from older age groups (more than 55 years) reported a slight to a significant decrease in the mentioned travel outcomes.
- *Years of smartphone use*: It was observed that the respondents who are more experienced with using smartphones showed more changes in travel outcomes due to smartphone app usage. All respondents with less than one year of experience showed no impact on VKT for work/education. Even with one to three years of experience, only 2% of respondents stated a slight increase, and with three to five years of experience, only 8% stated a slight increase, and 1% stated a significant increase in VKT. With more than five years of experience with smartphone usage, 25% stated a slight increase, and 13% stated a significant increase in VKT for work/education trips. Nobody stated a decrease in VKT for the same with a gain of smartphone usage experience. In the

case of VKT for shopping trips, all respondents with less than one year of experience stated no impact. However, unlike VKT for work/education trips, just with one to three years of experience, 20% of respondents stated a slight decrease, and with three to five years of experience, 34% of respondents stated a slight to significant decrease in VKT due to app usage. About 75% of respondents stated a slight to significant decrease in VKT with more than five years of experience with smartphone use. No increase was stated for this outcome. Another observation was that smartphone app users with less experience reported significant changes in VKT for recreation trips and other recreation-based outcomes. Forty percent of respondents with less than one year of experience reported a slight decrease in VKT for recreational trips. Although most respondents indicated no impact, 30%, 40%, and 30% of respondents with less usage experience reported a significant decline in the number of social gatherings attended, new places visited, and group trips planned, respectively, because of smartphone app usage. However, more experienced users for these purposes reported both rises and declines, with the decrease slightly greater than the increase in VKT for recreation, social gatherings attended, new places visited, and group trips planned.

Among the factors taken into consideration at the household level, the household composition shows a statistically significant correlation with some travel outcomes, such as the frequency of social gatherings, new places visited, and scheduled group trips (with a chi-square test at a 5% significance level). Table 5 shows the household-level impacts of smartphone apps usage on travel outcomes.

**Table 5.** Household-level attributes and travel outcome-wise classification of smartphone app users.

| Travel Outcome | Stated Impacts | HH Composition | | Monthly HH Income | | | | | Vehicle Ownership | | | | | | | |
| | | | | | | | | | Four-Wheeler | | | | Two-Wheeler | | | |
| | | With Children | No Children | <₹ 5 k | ₹ 5 k–20 k | ₹ 20 k–50 k | ₹ 50 k–100 k | >₹ 100 k | None | One | Two | >2 | None | One | Two | >2 |
|---|---|---|---|---|---|---|---|---|---|---|---|---|---|---|---|---|
| VKT for Work/Education | Significant Decrease | 0% | 0% | 0% | 0% | 0% | 0% | 0% | 0% | 0% | 0% | 0% | 0% | 0% | 0% | 0% |
| | Slight Decrease | 0% | 0% | 0% | 0% | 0% | 0% | 0% | 0% | 0% | 0% | 0% | 0% | 0% | 0% | 0% |
| | No Impact | 69% | 77% | 100% | 96% | 68% | 56% | 43% | 95% | 64% | 44% | 48% | 94% | 72% | 61% | 31% |
| | Slight Increase | 22% | 15% | 0% | 4% | 27% | 24% | 37% | 5% | 27% | 31% | 33% | 5% | 21% | 25% | 34% |
| | Significant Increase | 9% | 8% | 0% | 0% | 4% | 20% | 20% | 1% | 9% | 25% | 18% | 1% | 7% | 13% | 34% |
| | | *p*-value = 0.139 * | | *p*-value = 0.000 | | | | | *p*-value = 0.000 | | | | *p*-value = 0.000 | | | |
| VKT for Shopping Trips | Significant Decrease | 40% | 32% | 0% | 5% | 55% | 64% | 56% | 8% | 53% | 62% | 61% | 10% | 33% | 61% | 62% |
| | Slight Decrease | 22% | 25% | 0% | 38% | 32% | 25% | 22% | 18% | 32% | 22% | 15% | 11% | 31% | 24% | 24% |
| | No Impact | 38% | 43% | 100% | 57% | 14% | 11% | 22% | 75% | 14% | 16% | 24% | 79% | 36% | 15% | 14% |
| | Slight Increase | 0% | 0% | 0% | 0% | 0% | 0% | 0% | 0% | 0% | 0% | 0% | 0% | 0% | 0% | 0% |
| | Significant Increase | 0% | 0% | 0% | 0% | 0% | 0% | 0% | 0% | 0% | 0% | 0% | 0% | 0% | 0% | 0% |
| | | *p*-value = 0.211 * | | *p*-value = 0.000 | | | | | *p*-value = 0.000 | | | | *p*-value = 0.000 | | | |
| VKT for Recreational Trips | Significant Decrease | 5% | 5% | 0% | 0% | 8% | 4% | 11% | 0% | 7% | 7% | 12% | 0% | 5% | 10% | 3% |
| | Slight Decrease | 21% | 23% | 21% | 25% | 24% | 28% | 11% | 22% | 25% | 20% | 12% | 20% | 27% | 19% | 10% |
| | No Impact | 40% | 44% | 79% | 75% | 19% | 19% | 17% | 73% | 20% | 20% | 12% | 72% | 39% | 21% | 17% |
| | Slight Increase | 25% | 20% | 0% | 0% | 48% | 32% | 34% | 5% | 38% | 28% | 42% | 8% | 22% | 37% | 31% |
| | Significant Increase | 9% | 9% | 0% | 0% | 0% | 17% | 28% | 0% | 10% | 25% | 21% | 0% | 8% | 13% | 38% |
| | | *p*-value = 0.592 * | | *p*-value = 0.000 | | | | | *p*-value = 0.000 | | | | *p*-value = 0.000 | | | |
| Number of Social Gatherings | Significant Decrease | 3% | 13% | 11% | 6% | 8% | 8% | 5% | 8% | 8% | 7% | 6% | 10% | 7% | 9% | 0% |
| | Slight Decrease | 25% | 20% | 22% | 29% | 22% | 24% | 18% | 25% | 22% | 25% | 15% | 27% | 21% | 25% | 17% |
| | No Impact | 49% | 46% | 67% | 64% | 39% | 35% | 34% | 64% | 37% | 31% | 42% | 57% | 52% | 35% | 34% |
| | Slight Increase | 16% | 15% | 0% | 0% | 31% | 24% | 25% | 4% | 27% | 22% | 21% | 6% | 16% | 25% | 24% |
| | Significant Increase | 6% | 5% | 0% | 0% | 0% | 8% | 18% | 0% | 5% | 15% | 15% | 0% | 5% | 7% | 24% |
| | | *p*-value = 0.002 | | *p*-value = 0.000 | | | | | *p*-value = 0.000 | | | | *p*-value = 0.000 | | | |
| Number of New Places Visited | Significant Decrease | 6% | 14% | 13% | 11% | 11% | 8% | 5% | 11% | 10% | 6% | 6% | 12% | 8% | 10% | 7% |
| | Slight Decrease | 19% | 22% | 21% | 36% | 13% | 19% | 14% | 27% | 15% | 16% | 18% | 25% | 23% | 15% | 10% |
| | No Impact | 49% | 47% | 66% | 54% | 38% | 41% | 40% | 60% | 39% | 36% | 36% | 60% | 45% | 41% | 45% |
| | Slight Increase | 20% | 14% | 0% | 0% | 39% | 22% | 26% | 3% | 32% | 22% | 27% | 4% | 19% | 26% | 28% |
| | Significant Increase | 6% | 4% | 0% | 0% | 0% | 9% | 15% | 0% | 5% | 14% | 12% | 0% | 5% | 8% | 10% |
| | | *p*-value = 0.018 | | *p*-value = 0.000 | | | | | *p*-value = 0.000 | | | | *p*-value = 0.000 | | | |
| Number of Group Trips Planned | Significant Decrease | 5% | 13% | 9% | 7% | 11% | 11% | 7% | 8% | 10% | 10% | 9% | 10% | 7% | 14% | 0% |
| | Slight Decrease | 19% | 22% | 17% | 32% | 19% | 21% | 14% | 23% | 21% | 16% | 12% | 23% | 23% | 13% | 21% |
| | No Impact | 54% | 46% | 74% | 61% | 40% | 44% | 33% | 66% | 39% | 47% | 21% | 63% | 52% | 40% | 28% |
| | Slight Increase | 16% | 14% | 0% | 0% | 31% | 15% | 28% | 3% | 24% | 15% | 39% | 5% | 14% | 21% | 34% |
| | Significant Increase | 5% | 5% | 0% | 0% | 0% | 9% | 18% | 0% | 6% | 12% | 18% | 0% | 4% | 10% | 17% |
| | | *p*-value = 0.040 | | *p*-value = 0.000 | | | | | *p*-value = 0.000 | | | | *p*-value = 0.000 | | | |

*: Null hypothesis accepted.

Impact of smartphone app usage at the household level on travel outcomes for statistically significant relations are discussed below:

- *Household Composition*: A large proportion of respondents from both types of households stated no impact on travel outcomes. For stated changes in VKT (for work/

education trips, shopping trips & recreational trips) higher proportion of respondents from households without children have stated no impact as compared to the ones with children. For the remaining outcomes, higher proportion of respondents from households with children have stated no impact. For stated changes in VKT for work/education, nobody stated any decrease and for the same outcome, 31% from households with children stated slight to significant increase while only 23% from households without children stated the same. For stated changes in VKT for shopping, nobody stated any increase and for the same outcome, 62% from households with children stated slight to significant increase while 57% from households without children stated the same. For stated changes in VKT for recreational trips, 26% from households with children and 27% from households without children stated slight to significant decrease while 34% from households with children and 29% from households without children stated slight to significant increase. For stated change in number of social gatherings attended, 28% from households with children and 33% from households without children stated slight to significant decrease while 22% from households with children and 20% from households without children stated slight to significant increase. For stated change in Number of New Places Visited, 25% from households with children and 36% from households without children stated slight to significant decrease while 26% from households with children and 18% from households without children stated slight to significant increase. For stated change in Number of Group Trips Planned, 24% from households with children and 35% from households without children stated slight to significant decrease while 21% from households with children and 19% from households without children stated slight to significant increase.

- *Monthly Household Income*: None of the respondents with a household income of less than INR 5000 stated any change in VKT for work/education, but with the increase in income levels, respondents stated a slight to significant increase in VKT for the same. No respondent stated any decrease in VKT for work/education because of smartphone app usage. For VKT for shopping trips, the number of respondents stating no impact became smaller with an increase in income, and respondents reported a slight to significant decrease in VKT for the same outcome. An exception is a group with an income of more than INR 100,000 where respondents stated no change (22%), although still very low, it was higher than the relatively low-income group (INR 50,000 to 100,000) and thus, the proportion of respondents stating change was also comparatively lower. No respondent stated an increase in VKT for shopping trips because of smartphone app usage. In the case of VKT for recreational trips, it was observed that unlike VKT for the other two outcomes, here, even the lowest income group stated a slight decrease (21%). As the income increased, the proportion of respondents stating a decrease in VKT and those reporting an increase in VKT increased, so much so that in the highest income category, most respondents (62%) stated an increase in VKT because of app usage. A similar trend was observed for other outcomes, such as the number of social gatherings attended, new places visited, and group trips planned, with an even more significant proportion of respondents from the lower income categories who stated a slight to significant decrease and a little lower proportion of respondents from the higher income categories who stated a slight to significant increase in travel outcomes.

- *Vehicle Ownership*: It was observed that most users with no household vehicles (either four-wheeler or two-wheeler) stated no impact on all travel outcomes, especially VKT for work/education (95%), and as the number of vehicles increased, the users reported changes in travel outcomes. No decrease in VKT for work/education was stated, but with the increase in the number of vehicles owned, a slight to significant increase was stated in most households with exactly two four-wheelers (56%) and households with more than two two-wheelers (68%). In the case of VKT for shopping trips, no increase was stated, and respondents from households with fewer vehicles (one to two four-wheelers or two-wheelers) stated a slight to a significant decrease in shopping

trips, especially if the household had two four-wheelers (84%) or if they had two or more two-wheelers (86%). In the case of VKT for recreational trips, as the number of vehicles owned increased, changes were stated, and the proportions of respondents who stated a slight to significant increase in VKT in each income category were much larger than those who stated a decrease in VKT. For outcomes, such as the number of social gatherings, new places visited, and group trips planned, most users from households with a greater number of vehicles owned reported a slight to significant increase in travel and vice-versa.

## 5. Discussion of Results

In this section, the key findings of the study are presented across various socio-demographic classifications.

### 5.1. App Usage for Trip Planning Purposes

It was observed that most respondents across socio-demographic classifications showed a relatively higher dependence on smartphone apps for communication and coordination with co-travellers, which is consistent with previous findings [13]. Moreover, when choosing a mode of transport, most respondents in Bhopal showed the least dependence across all classifications, which has not previously been observed. When it came to deciding when to leave, male respondents were slightly more reliant on smartphone apps, whereas women used them more for selecting a mode of transportation. So, a slight difference in app usage for some trip planning purposes was observed in Bhopal across genders, which has also been observed before [28]. However, for all other trip planning purposes, app usage was observed to be gender neutral, as also previously observed [4]. Generally, younger users were found to be more dependent on app usage for all trip planning purposes, which has also been previously observed [13,28], with the exception of the purpose of deciding when to depart for which, users between ages 25 to 54 years showed a higher dependence on smartphone apps in Bhopal. Considering the number of years that users owned a smartphone, as their experience with ownership increased, respondents tended to show a significantly higher dependence on app usage for all purposes. This is slightly different from the findings available [13] where the dependence on apps has been found to increase very gradually.

At the household level, it was observed that the dependence on smartphone apps had not been affected by the presence of children under 18 years in households in Bhopal, except for the purpose of deciding when to depart for various activities on a daily basis, considering the routine or children's schedule. The dependence on app usage by respondents from high-income households was observed to be higher for all purposes and steadily decreased for lower-income households, which is consistent with previous studies [11,28,29]. Although the dependence on smartphone apps for choosing a mode of transport was found to be the least of all purposes, it was observed that the users of medium-income households had a relatively higher app usage for the same. Surprisingly, unlike the research evidence [13], apart from the purpose of choosing a mode of transport, households that did not own a vehicle showed a lower dependence on app usage and vice-versa. It was observed that users among families with at least one private car appeared to demonstrate more reliance on smartphone applications for various trip planning purposes.

### 5.2. Stated Changes in Travel Outcomes

The majority of respondents stated no changes in travel outcomes because of app usage for trip planning, which is consistent with the available research, except for users between the ages of 18 to 34 years in Bhopal, who reported significant impacts. Findings specific to stated changes in VKT for work/education and recreational trips in this study are unique and these have not been attempted before. Additionally, findings specific to the number of social gatherings, the number of new places visited, and the number of group trips planned are also very limited and only one such attempt was made earlier [13] where

app usage across age, vehicle ownership, and transit pass ownership were found to have a significant impact on travel outcomes. In contrast, this study for Bhopal studied such a relationship across a wider spectrum of significant socio-demographic classifications.

For VKT for work/education, no respondents across classifications stated any decrease. A larger proportion of male respondents stated an increase in this travel outcome. Changes in VKT for work/education was found to be inversely proportional to the age of the respondent and thus, a larger proportion of younger users stated an increase. A larger number of respondents with more years of experience with smartphone app usage for trip planning activities stated an increase. A rise in income was also found to be a significant factor and it was observed that a larger proportion of respondents from high-income households stated a slight to significant increase. A rise in income also encouraged ownership of both smartphone devices and vehicles (two- and four-wheelers), and a larger proportion of respondents from households with a greater number of vehicles stated a slight to significant increase.

For VKT for shopping trips, the findings are consistent with some of the available literature [11], as no respondents across classifications stated any decrease in this outcome. A slightly larger proportion of female respondents stated a decrease because of a dependence on app usage for trip planning purposes. A larger proportion of younger users stated a decrease. Respondents with more years of experience with smartphone app usage for trip planning activities stated a decrease in this outcome. A significantly larger proportion of respondents from high-income households stated a slight to significant decrease. The majority of respondents from households with a greater number of vehicles stated a slight to significant decrease.

A significantly large share of respondents stated no changes in VKT for recreational trips. Among those who reported a change, most of the younger respondents reported a slight to significant increase. However, as age increased, more and more users stated that the number of recreation trips decreased. Users with less experience with smartphones reported a decrease and vice-versa. As income increased, a larger share of respondents stated an increase in VKT, rather than a decrease. A similar finding was observed in the case of vehicle ownership.

As with VKT for recreational trips, a significantly large share of respondents stated no changes in the number of social gatherings, the number of new places visited, and the number of group trips planned because of a dependence on smartphone apps. Among those who reported a change, most of the younger respondents reported a slight to significant increase. However, as age increased, more and more users stated a decrease in these outcomes. The findings of these travel outcomes, specific to users under 35 years of age are consistent with the findings in the available literature [13] but vary significantly for older users. Respondents with less experience in using smartphones reported a decrease in these outcomes due to app usage, whereas those with more experience reported an increase. Since low-income households also have fewer vehicles, the usage of trip apps seems to encourage a decrease in these activities and vice-versa. This is also a unique finding of this research, as pre-dominantly no impact has been observed in the available literature [13].

## 6. Conclusions

In context of Indian cities, there is close to no knowledge about how ICT-based smartphone apps meant for day-to-day trip planning tasks influence travel outcomes. So, an attempt has been made in this study to better understand the patterns of smartphone app usage for daily trip planning purposes, and changes in travel outcomes because of app usage using an exploratory analysis of a sample of respondents of Bhopal. The chi-square test has been used to establish if the differences in the app usage patterns and stated changes in travel outcomes of the respondents across socio-demographic classification occur due to an inherent relationship between those socio-economic factors and the frequency of app usage or even the travel outcomes.

Similar to the existing literature, it has been found that most respondents use the smartphone apps for the purpose of communication or coordination. A slight difference in app usage for some trip planning purposes across genders has also been observed. Male respondents are more dependent on apps for deciding when to leave whereas, female respondents are more dependent on apps for selecting a mode of transportation. Apart from these two purposes, app usage has been found to be gender neutral. Younger users have been found to be more dependent on app usage for most trip planning purposes. The dependence on app usage by respondents from high-income households has been observed to be higher for all purposes and steadily decreases for lower-income households. The majority of respondent stated no changes in travel outcomes because of app usage for trip planning purposes. Furthermore, no respondents across classifications stated any decrease in VKT for shopping trips. Most of the younger respondents reported a slight to significant increase. Most of the younger respondents reported a slight to significant increase in the number of social gatherings, the number of new places visited, and the number of group trips planned because of app usage for trip planning purposes.

Many findings that are unique to the context of the study area have also been observed. For instance, respondents in Bhopal have shown the least dependence on app usage across all classifications when choosing a mode of transport. For the purpose of deciding when to depart, users between ages 25 to 54 years showed a higher dependence on app usage. With the increase in experience with smartphone ownership, a significantly higher dependence on app usage for all purposes has been observed. The dependence on smartphone apps has not been affected by the presence of children under 18 years of age, except for the purpose of deciding when to depart. Although the dependence on smartphone apps for choosing a mode of transport has been found to be the least of all purposes, medium-income households have comparatively higher app usage for the same. Apart from the purpose of choosing a mode of transport, households which do not own a vehicle have shown a lower dependence on app usage and vice-versa. All of the findings specific to stated changes in VKT for work/education and recreational trips in this study are also unique. For VKT for work/education, no respondents across classifications stated any decrease. A significantly large share of respondents stated no changes in VKT for recreational trips. Respondents with less experience in using smartphones reported a decrease in trip making for social gatherings, visiting new places, and group trips due to app usage, whereas those with more experience reported an increase.

Although most respondents in the study rejected the substitution or diminishing effects of smartphone use on travel, it will be interesting to further investigate how socio-economic and other factors affect the use of smartphone apps for various travel-related purposes, and their effects on travel outcomes through a multivariate analysis, which will highlight the trade-offs between different factors. Finding latent influences on this relationship between smartphone app usage and travel behaviour would be interesting as well and might help with the introduction of new app-based mobility models, such as MaaS, for offering effective mobility solutions tailored to the needs of different user groups. A more thorough analysis of the factors influencing mobility app users' transportation choices in other Indian metropolises and cities would produce more useful results and aid in the development of better policies to encourage the use of new mobility services. Additionally, a detailed study on the mobility of children, their dependence on ICT, and their role in bridging the digital divide within the households is needed.

Nevertheless, the study's findings show that the smartphone apps need to be more user-specific and personalised based on socioeconomic/demographic criteria to offer consumers customised mobility alternatives that match their demands. The outcomes of this study also provide important behavioural insights. They may be helpful in policy discussions regarding the inception of a smartphone app-based mobility ecosystem in a medium-sized city like Bhopal, collecting the necessary data, and conducting the mentioned multivariate studies for aiding its deployment. Public organisations, regional communities, and app developers will need to collaborate on this. To help build a connected community

platform where residents can use smartphone apps for their travel decisions, planners should propose high quality and affordable facilities, such as free wi-fi at all places of interest (e.g., bus stops, recreation areas, walking paths, shopping centres, etc.).

**Author Contributions:** Conceptualization, S.G. and K.S.; methodology, S.G. and K.S.; software, K.S.; validation, S.G. and K.S.; formal analysis, K.S.; investigation, K.S.; resources, K.S.; data curation, K.S.; writing—original draft preparation, K.S.; writing—review and editing, K.S. and S.G.; visualization, K.S.; supervision, S.G. All authors have read and agreed to the published version of the manuscript.

**Funding:** This research received no external funding.

**Institutional Review Board Statement:** Not Applicable.

**Informed Consent Statement:** Informed consent was obtained from all subjects involved in the study.

**Data Availability Statement:** The data that are presented in this study are available upon request from the corresponding author. The data are not publicly available as they are part of an ongoing doctoral research.

**Conflicts of Interest:** The authors declare no conflict of interest.

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
