# Peer review of "Smartphone App Usage Patterns for Trip Planning Purposes and Stated Impacts in the City of Bhopal, India"

_urbansci, doi:10.3390/urbansci7010025_

Round 1

Reviewer 1 Report

The study examines the mobile app use and trip plan effects in India. While the study offers some insights, there are key some key aspects to improve the quality of the paper.

Intro: Rather than delving into the India case, the author(s) should provide some general information on the subject.

Since the study did not include individuals who are below 18, this may create a major problem for the study purpose (socio-demographic aspects of App use etc.)

What are the survey questions?

What App information did the author(s) collect? What was the motivation for it?

There are some cliche findings/ or already known facts on this context. For instance, young generation tend to use more App etc. So, these are not adding new information indeed.

Since old generation tend to use less mobile phone, more specifically Apps, the author(s) should seek for some key findings/approaches.

How did the author(s) approach individuals who do not use App but use such point of interests?

Reviewer 2 Report

A survey of smartphone users was conducted online to gather the primary data for 157 this study. - Please explain more about the process of collecting data

With data analysis, please show demographic information in the table.

Reviewer 3 Report

This study reports an analysis of data on the use of smartphone app in travel, based on data from Bhopal, India. There are various efforts worldwide to use integrated smartphone applications for travel, and how to analyze and leverage such efforts could be an important research question. 

However, this study needs to be significantly revised as an academic paper.

From a major perspective, issues in extant literature have not been clarified sufficiently. The authors should clarify what research gap this study addresses by presenting the extant literature. 

The second point is an explanation of the validity of adopting the analytical method. The chi-square test only tests whether there is a relationship between two variables. The authors should clearly explain what extended perspectives can be offered on existing theories from this perspective.

Below are comments tied to the line numbers.

-------------------------

# Line 10: Abstract

The articulation of research gaps is an important aspect of academic research in the social sciences. In the abstract, the authors should provide an explanation that reads the research gap addressed in this study. The research gap is the boundary between perspectives that have already been articulated in the extant literature and those that have yet to be articulated.

# Line 16: Abstract

"se-lecting" should be "selecting". 

# Line 20: Abstract

"in-fluence" should be "in-fluence". 

# Line 83: 1. Introduction, Par.8

The authors should explain what the research gap this study establishes is before presenting the research purpose. The research gap is the boundary between perspectives that have already been articulated in the extant literature and those that have yet to be articulated. Please present the overview in the Introduction section, and explain the details in the Literature Review section.

# Line 83: 1. Introduction, Par.8

The authors should also explain the prospects of what contribution can be made by the achievement of the objectives of this study. This perspective will clarify the scope to which this study can be adapted.

# Line 94: 2. Literature Review, Par.2:

"neutral.[12]." should be "neutral [12].". 

# Line 121: 2. Literature Review, Par.3:

"at-tended" should be "attended". 

# Line 123: 2. Literature Review

In the Literature Review section, authors should provide an explanation of perspectives on what points have not been clarified and how to position this study against them, in addition to the insights that have been clarified in the extant literature so far.

# Line 124: 3.1 Case Study

I understand that a case study is a qualitative research method that requires a detailed case description based on interviews, documents, and so on. The appropriateness of applying this research method should also be explained by presenting relevant literature on the methodology.

# Line 163: 3.2. Data Collection, Par.1

"in-cluding" should be "including". 

# Line 170: 3.2. Data Collection, Par.2

Authors should explain how previous studies have discussed the indicators for the questions used in this survey. If new indicators need to be developed for this study, please provide an explanation to ensure the validity and reliability. 

# Line 185: 3.3. Data Analysis, Par.1

For academic research, the major focus is on identifying the relationship behind the data. The chi-square test can only test for the existence of an association, but please also explain directions on how existing theory can be extended by confirming the existence of such a relationship with the chi-square test. 

# Line 215: Table 2. 

In the chart/table, authors should present information that can be read from the chart/table alone. For example, in Table 2, it is not possible to read from the table alone what the numbers presented mean.

# Line 274: Figure 3. 

It appears that Figure 3 is not referenced in the text. Please be sure to refer to the figures and tables in the text.

# Line 420: 4. Results and Discussion

There are three main general perspectives that need to be discussed in business research in the social sciences: Novelty of results, theoretical implications, and practical implications. It should be structured in such a way that each perspective can be clearly read.

# Line 426: 5. Conclusion

"be-have" should be "behave". 

-------------------------

Round 2

Reviewer 1 Report

Thank you for addressing my comments.

Author Response

No additional comments/suggestions have been provided.